# Optimization of MIMU Mounting Position on Shank in Posture Estimation Considering Muscle Protuberance

**DOI:** 10.3390/s25072273

**Published:** 2025-04-03

**Authors:** Shun Kanega, Yoshihiro Muraoka

**Affiliations:** 1Faculty of Human Sciences, Waseda University, Tokorozawa 359-1192, Japan; shunrun@fuji.waseda.jp; 2National Hospital Organization Murayama Medical Center, Tokyo 208-0011, Japan

**Keywords:** IMU, MIMU, optimal position, shank, gait, pacemaker, muscle contraction, soft tissue artifact

## Abstract

The influence of the mounting position of a magnetic-inertial measurement unit (MIMU) on the accuracy of posture estimation for a shank has not been extensively studied and remains unknown. In this study, we conducted comparative experiments using three MIMU positions: the lateral and frontal positions, which are commonly used, and the medial tibial position, which is less affected by muscle protuberance, considering the anatomical structure of the body. To determine the optimal MIMU mounting position on the shank, we repeatedly performed plantar–dorsiflexion and relaxation of the ankle joint in a chair-sitting position and examined the effect of muscle contraction on the posture of the MIMU (Experiment 1). We also performed posture estimation during gait and compared the three-dimensional shank posture measured by the MIMU and optical motion capture to evaluate the estimation accuracy for each mounting position (Experiment 2). In Experiment 1, the orientation change at the medial tibia was significantly smaller than that at the other positions, showing an 80% reduction compared with the anterior tibia during dorsiflexion. In Experiment 2, the medial tibia achieved the highest estimation accuracy, showing a 13% lower RMSE than that of the anterior position. The results of these two experiments suggest that the medial tibia is the optimal position on the shank, as the posture estimation accuracy was the highest when the MIMU was mounted on the medial tibia, where there was no muscle under the mounting surface. Moreover, the posture estimation accuracy was less affected by muscle protuberance under these conditions.

## 1. Introduction

### 1.1. Gait Analysis and Monitoring with MIMU

Gait analysis provides useful information regarding the process from diagnosis to rehabilitation and discriminates between normal and abnormal gait [1,2,3], determines the level of motor function [4], and evaluates the effects of rehabilitation interventions [5]. For example, the degree of deviation from normal gait can be quantified using spatiotemporal gait parameters, such as gait speed, stride length, joint angle, shank tilt angle, and stance and swing times. Moreover, changes in gait over time can be recorded quantitatively [1]. In clinical practice, gait analyses are performed by therapists and physicians because they are simple and inexpensive. However, subjective evaluations based on observation are unreliable. Previous studies have indicated that it is particularly difficult to accurately capture the joint and posture angles during gait [6,7], and the acquisition of temporal gait parameters is unreliable. However, quantitative and accurate optical motion capture (MOCAP) is the gold standard in laboratories and gait analysis rooms and is frequently used in kinematic analyses of the lower limbs [8]. However, these methods require expensive equipment and a dedicated measurement environment, which limits their applicability. Another major limitation is the time-consuming preparation process that requires affixing multiple reflective markers to the body.

Under these circumstances, many posture estimation methods using inertial measurement units (IMUs) have been proposed to perform gait analysis more easily and at a lower cost [9,10,11,12,13,14]. IMUs are integrated with acceleration and angular rate sensors. Those with additional magnetic sensors are called magnetic-inertial measurement units (MIMUs) or magnetic, angular rate, and gravity units (MARGs). Wearable MIMUs have advantages, including their low cost, small size, and light weight, and they can be easily mounted and detached from the body, making them suitable for clinical applications. These advantages have made it possible to perform gait analysis not only in a laboratory environment but also at home and in everyday life. The MIMU also enables the monitoring and evaluation of various diseases and movement analyses in daily life, including the evaluation and monitoring of joint motion and gait symmetry after total knee replacement surgery [15,16], gait after a spinal cord injury [17], Parkinson’s disease [18], and gait balance after a stroke [19]. Furthermore, mounting a MIMU on the shank enables gait phase estimation, which can be used for not only gait analysis and evaluation but also the posture control of prosthetic limbs [20] and exoskeletal robots [21], control of functional electrical stimulation (FES) [22], selection of orthotics and short-leg orthotics [23], and adjustment of the braking forces of short-limb orthotics. For example, in stroke hemiplegics, FES is used in the lower limbs to support the functional decline of the paralyzed limb [24]. Thus, information on posture angles, such as the shank tilt angle and gait phase, is necessary for timing control. In addition, short-leg orthoses are often used to promote heel grounding and utilize the rocker function in hemiplegics, and the braking force and initial angle of the orthosis must be adjusted according to the shank-to-vertical angle (SVA) [25].

### 1.2. Review of Previous Studies on MIMU Mounting Positions and Issues

Many previous studies have estimated the posture and joint angles of the lower limbs by mounting MIMUs on various body parts; however, few studies have examined the appropriate mounting position of the MIMU. Moreover, the effect of the mounting position of the MIMU on the accuracy of the posture estimation is unknown. Table 1 shows examples of MIMU mounting positions on the shank in previous studies [26,27,28,29,30,31,32,33,34,35,36,37,38,39] that conducted posture estimations of the lower limbs. Several previous studies mounted sensors on the front or outside of the shank along the body axes in the sagittal and frontal planes [26,27,28,29,30,31,32,33,34,35,36]. However, some studies mounted sensors on the medial tibia, although the number of references was small compared with those at other positions [37,38,39]. Thus, the position of the MIMU varies, and no standard position has been defined.

Currently, few studies have examined MIMU mounting positions; however, these studies examined combinations of MIMU mounting positions during gait and squat movements [40,41]. Wesley et al. (2020) estimated knee joint angles during a timed up-and-go (TUG) task by mounting MIMUs at positions commonly reported in previous studies (four thigh and three shank positions) and examining combinations of thigh and shank MIMU mounting positions [40]. They reported that accurate estimates of knee joint flexion/extension could be obtained when the MIMU was mounted in combinations of the lower front thigh and outer mid-shank, outer mid-thigh and outer mid-shank, and lower front thigh and outer lower shank. However, this study was limited to the sagittal plane kinematics. Mathias et al. (2023) studied the positions of IMUs on the thigh during squatting [41]. Three IMUs were mounted on the lateral aspect of the thigh (IMU1: midpoint between the greater trochanter and lateral femoral condyle; IMU2: 2 cm proximal to IMU1 on the same axis; IMU3: 2 cm distal to IMU1 on the same axis). The results showed that the estimation error was the smallest at the position closest to the knee joint (IMU3). In addition to posture estimation, previous studies examined the MIMU mounting position in the measurement of gait parameters, such as the stance/swing ratio and cadence, and in the detection of gait events [42,43]. Panebianco et al. (2018) compared the accuracy of gait event detection when IMUs were mounted on the trunk, foot, and shank, and they reported that algorithms based on shank and foot IMUs were more accurate and reproducible in gait event detection than were algorithms based on torso IMUs [42]. Anwary et al. (2018) studied the optimal foot mounting position for gait event detection and concluded that the medial side of the foot (above the first cuneiform process) was optimal for ensuring sensor stability [43].

Thus, all investigations of the MIMU mounting positions were based on kinematic results. However, the body on which the MIMU is mounted is not perfectly rigid, and the shape of the body segment changes because of muscle contraction and shortening. When the sensor is placed on or near a muscle, muscle contraction causes the expansion and elevation of the mounting surface (muscular protuberance). Posture estimation is easily affected by the orientation change of the MIMU. As shown in Figure 1, the posture of the MIMU changes when the mounting surface is raised, which results in large errors. Therefore, in addition to kinematic results, such as posture estimation error compared with motion capture, it is necessary to consider the MIMU mounting position with respect to the anatomical body structure and the effects of muscle protuberance owing to muscle contraction and shortening.

### 1.3. Muscular Protuberance and MIMU Mounting Position

Unlike robots and prostheses, the body does not have perfectly flat or perpendicular areas. Therefore, the MIMU mounting position was variable. Although previous studies tended to mount sensors along the body axis, such as on the sides and front [26,27,28,29,30,31,32,33,34,35,36], the body is not rigid, and the shape of the body surface changes with muscle contraction and shortening. For example, the root mean square error (RMSE) of the knee joint flexion/extension angle was reported to be less than 1° for a prosthetic leg and approximately 3° for a human leg [44]. As shown in Figure 1, during relaxation, the muscle thickness was small, and the MIMU surface was relatively flat with a small curvature. However, during contraction, the muscle thickness increases, and the surface of the MIMU becomes more curved than that during relaxation (muscular protuberance). Consequently, the MIMU tilts and posture fluctuates, which is thought to cause posture estimation errors owing to muscle contraction and shortening. However, the extent to which these factors affect the accuracy of posture estimation using MIMU remains unclear. In addition to the effects of muscle expansion, soft tissue artifacts (STAs), including skin displacement and soft tissue deformation, are widely recognized as major sources of error in human motion analyses. STAs result from the relative motion between skin, muscle, and underlying bone during dynamic activity and can significantly distort kinematic measurements. Potentially, motion capture STA can be a major source of distortion of kinematic measurements. Several studies have quantified the STA when optical motion capture markers are attached to the body and have shown that they introduce inaccuracies in kinematic calculations [45,46,47,48,49]. In addition, compensation methods have been proposed for optical motion capture systems [50,51]; however, such methods are not applicable to wearable IMU-based cases. Thus, a more realistic and clinically relevant approach to gait analysis using an IMU would be to minimize the effects of the STA by selecting an anatomically stable mounting position, such as the medial tibia. 

Therefore, it is necessary to clarify how soft tissue artifacts, such as muscle protrusions, affect the estimation of lower limb kinematics, depending on the position of the MIMU, and to evaluate the optimal sensor mounting position, considering the anatomical body structure. In this study, we compared three MIMU mounting positions on the shank: lateral and anterior, which are common mounting positions, and the medial tibia, which is less affected by muscle protuberance when the anatomical body structure is considered. Two experiments were conducted to determine the optimal MIMU mounting position on the shank. The first experiment investigated the effect of muscle contraction on the MIMU orientation by repeatedly performing plantar dorsiflexion and relaxation of the ankle joint in a chair-sitting position. The second was an experiment to estimate posture during gait, compare the three-dimensional (3D) posture angles estimated by the MIMU with the posture angles measured via motion capture, and evaluate the estimation accuracy for each mounting position. Although many previous studies have focused on improving posture-estimation algorithms or comparing different sensor systems, the present study offers a novel perspective by comparing MIMU mounting positions under identical movement conditions and experimentally examining the influences of muscle contraction on the estimation error. This dual approach provides new insights into the optimization of sensor placement for MIMU-based gait analyses.

## 2. Materials and Methods

### 2.1. MIMU Mounting Positions

The positions of the MIMUs used in this study are shown in Figure 2. In addition to the common mounting positions of the lateral (MIMU1) and anterior shank (MIMU2), an MIMU was mounted on the medial tibia (MIMU3). The medial tibia was selected because there were no muscles under the mounting surface, and the orientation change of the MIMU owing to muscle protuberance was considered small. In contrast, the anterior surface of the shank was located above the tibialis anterior muscle, the lateral surface of the shank was located above the peroneus longus muscle near the soleus muscle, and the mounting surface increased because of muscle contraction and shortening. The MIMU was mounted on the back of the leg using Velcro tape (removable) and fixed to the shank using an elastic band. The elastic band was wrapped around the MIMU such that the upper end of the band was aligned with the lower end of the tibial tuberosity, using the tibial tuberosity as a landmark.

### 2.2. MIMU Sensor Characteristics and Controller

Three MIMUs (ICM-20948, TDK InvenSense, San Jose, CA, USA) capable of measuring the three-axis acceleration, angular velocity, and magnetic field were used. The sensor characteristics are listed in Table 2. The MIMU was controlled using a Portenta H7 Lite Connected (ABX0004, Arduino, Scarmagno, Italy) microcontroller board via SPI communication. Three SPI-level conversion modules (TXU0304, Texas Instruments, Dallas, TX, USA) were used to convert the voltage from 3.3 to 1.8 V for SPI communication. The Wi-Fi communication function of the microcontroller board was used to send and receive data between the microcontroller and PC. The power supply was provided by a 3.8 V lithium polymer battery. The total weight of the device, including that of the battery, was 72.4 g.

### 2.3. Posture Estimation Using MIMUs

The posture estimation algorithm used in this study is shown in Figure 3. The Madgwick filter is a quaternion-based posture estimation filter that uses a gradient descent algorithm to fuse the acceleration, angular velocity, and magnetic estimates of orientation with a built-in magnetic distortion correction. The gain of the Madgwick filter was set to 0.1. The coordinate system is right-handed. The Earth frame is E (X_E_, Y_E_, Z_E_), where the negative direction of the *z*-axis is the direction of gravity and the positive direction of the *x*-axis is north. The body segment frame of the shank is B (X_B_, Y_B_, Z_B_), where the positive direction of the *X*-axis is rightward and the positive direction of the *Y*-axis is forward (Figure 2). The sensor frame representing the MIMU posture is S (X_S_, Y_S_, Z_S_), where the XY-plane is the mounting surface (Figure 2). The angle around the *X*-axis is the roll angle ϕ, that around the *Y*-axis is the pitch angle θ, and that around the *z*-axis is the yaw angle ψ. The YZ, XZ, and XY planes represent the sagittal, frontal, and horizontal planes, respectively. After calibrating the magnetic data, the three-axis acceleration, three-axis angular velocity, and corrected three-axis magnetic field were input into the Madgwick filter. The estimated posture quaternion of the MIMU was the output. To perform sensor-to-segment (S2S) calibration, the subjects were asked to stand still in a specified posture for a certain period, and the MIMU output data at rest were input to the Madgwick filter to calculate the MIMU, which was calculated by inputting the MIMU output data at rest into the Madgwick filter [52]. The posture quaternion of the MIMU output from the Madgwick filter was then converted into a body segment posture quaternion.

In 3D posture estimation, a magnetic sensor is required to correct the yaw angle (posture in the horizontal plane). However, the effects of magnetic disturbances caused by ferromagnetic materials present challenges. Because magnetic sensors are susceptible to the influences of surrounding ferromagnetic materials, such as the steel frames of buildings and the metals of medical equipment [53], magnetic calibration is important and requires recalibration each time a magnetic sensor is placed in a different environment. The relationship between the output value of the magnetic sensor and magnetic field is given by Equation (1). In this study, the sensitivity and offset values were calculated using ellipsoid fitting to correct the magnetic data. When the MIMU was rotated in any three-dimensional direction, ideal triaxial magnetic data without magnetic distortion were distributed on a sphere. However, the actual measured data were superimposed on the offset and distortion, as well as distributed on an ellipsoid. The data were fitted to an ellipsoid using the least-squares method and corrected to a sphere [54], where *m* is the magnetic sensor output, *M* is the corrected magnetic field, G is the sensitivity, and b is the offset.(1)$$M=Gm+b,\;M=\left[\begin{array}{c} M_{x}\\ M_{y}\\ M_{z} \end{array}\right],\;\;m=\left[\begin{array}{c} m_{x}\\ m_{y}\\ m_{z} \end{array}\right],\;\;G=\left[\begin{array}{ccc} G_{x_{1}} & G_{x_{2}} & G_{x_{3}}\\ G_{y_{1}} & G_{y_{2}} & G_{y_{3}}\\ G_{z_{1}} & G_{z_{2}} & G_{z_{3}} \end{array}\right],\;\;b=\left[\begin{array}{c} b_{x}\\ b_{y}\\ b_{z} \end{array}\right].$$

The quaternion (q) is used to represent the 3D posture, which is represented by a scalar ($q_{0}$) that represents the angle of rotation and a vector ($q_{1}$, $q_{2}$, $q_{3}$) that represents the axis of rotation. These relationships are expressed in Equation (2). Here, qES is a quaternion indicating the orientation of frame E relative to frame S. The conjugate quaternion is denoted by $q^{*}$ and can be used to swap the relative frames, as shown in Equation (3). For example, q*ES(qSE) is the conjugate quaternion of qES and represents the orientation of frame S relative to frame E.(2)qES=[ q0 q1 q2 q3 ],(3)q*ES=qSE=q0 −q1 −q2 −q3.

The quaternion qBE, which indicates the orientation of the body segment frame B relative to the Earth frame E, can be calculated using Equation (4). Here, qES is a quaternion indicating the orientation of the Earth frame E relative to the sensor frame S. Moreover, qES is the output from the Madgwick filter. Here, qBS is the quaternion of the body segment coordinate system relative to the sensor coordinate system and is the quaternion of the initial posture calculated from the stationary posture. The quaternion operation requires normalization by dividing the quaternion representing the posture by the norm, as expressed in Equation (5).(4)qBE=qSE ⨂  qBS =qES* ⨂ qBS ,(5)$$\left\|q\right\|=\sqrt{{q_{0}}^{2}+{q_{1}}^{2}+{q_{2}}^{2}+{q_{3}}^{2}}.$$

Quaternions were used in the Madgwick filter calculations, and the shank posture quaternions qBE were converted to Euler angles ϕ, θ, and ψ (rotation order: ZYX) [°] using Equation (6) to compare the final posture representation with the Euler angles measured by the motion capture. These posture estimation processes were performed using the Arduino IDE.(6)$$\left[\begin{array}{c} \phi \\ \theta \\ \psi \end{array}\right]=\left[\begin{array}{c} atan2(2{(q}_{2}q_{3}-2q_{0}q_{1},\;1-2{{(q}_{1}}^{2}+2{q_{2}}^{2}))\cdot 180/\pi \\ asin(-2\left(q_{1}q_{3}+2q_{0}q_{2}\right))\cdot 180/\pi \\ atan2(2{(q}_{1}q_{2}-2q_{0}q_{3},\;1-2{{(q}_{2}}^{2}+2{q_{3}}^{2}))\cdot 180/\pi \end{array}\right]$$

### 2.4. Subjects

The subjects were six healthy adults (three males and three females) without motor or sensory impairments of the lower limbs. All subjects were recruited from Waseda University. Table 3 presents the physical characteristics of the participants.

### 2.5. Experiment 1: Verifying the Effects of Muscle Contraction

#### 2.5.1. Objectives and Methods

The following experiment was conducted to verify the effects of muscle contraction on MIMU orientation. The postural changes in the MIMU were measured when the ankle joint was plantarflexed/dorsiflexed with the thigh and shank immobile and fixed in a chair-sitting posture, as shown in Figure 4. The shank was fixed with a homemade short-leg orthosis, and the thigh was firmly fixed with a non-stretchable belt. With the thigh and shank fixed, postural changes in the MIMU caused by changes in the shape of the body surface during muscle contraction and relaxation were measured. We measured the postural motion of the MIMU at the lateral and anterior shank and medial tibia under two conditions: dorsiflexion (up to approximately 20° dorsiflexion) with repeated relaxation and plantarflexion (up to approximately 20° plantarflexion) with repeated relaxation. Simultaneously, the EMG of the tibialis anterior muscle located on the anterior shank and the peroneus longus muscle located on the lateral shank were measured to monitor muscle activity during plantar–dorsiflexion, confirming that no muscle activity occurred during relaxation, the tibialis anterior muscle was active during dorsiflexion, and the peroneus longus muscle was active during plantarflexion. A two-channel wireless electromyograph (TS-MYO, Trunk Solution Corporation, Tokyo, Japan) was used for the EMG measurement of the shank and mounted on the belly of the target muscle with double-sided tape. The wireless electromyograph was controlled by a smartphone application and captured moving images synchronized with the EMG data. The sampling frequency was 1 kHz, and a bandpass filter was applied in the frequency range of 50–450 Hz. Three trials with three repetitions of plantarflexion, dorsiflexion, and relaxation were performed for each plantarflexion and dorsiflexion condition.

#### 2.5.2. Analysis (The MIMU Orientation Change)

The MIMU posture was calculated as the Euler angle in the sagittal, frontal, and horizontal planes, as described in Section 2.3. The absolute value of the maximum MIMU orientation change during plantar–dorsiflexion (MIMU orientation change) was calculated using the initial postural reference (0°) during chair-sitting relaxation. To examine the effects of muscle protrusions during the plantarflexion and dorsiflexion of the ankle joint, a repeated-measures two-way analysis of variance (ANOVA) was performed using the MIMU (lateral, anterior, and medial tibial) and motion (plantarflexion and dorsiflexion) as factors. The Bonferroni method was used for multiple comparisons, and the threshold for statistical significance for all tests was set as *p* < 0.05. R (ver. 4.2.2) was used for the analysis.

### 2.6. Experiment 2: Estimating 3D Shank Posture During Gait

#### 2.6.1. Gait Conditions

Three types of gait were performed: low-speed normal gait (slow, 2.0 km/h), medium-speed normal gait (medium, 3.0 km/h), and high-speed normal gait (fast, 4.0 km/h). Each walk was performed three times on level ground with bare feet. Gait speed is easy to control during treadmill gait but difficult to control during level gait. Therefore, to control the gait conditions (gait speed, stride length, and cadence), we fabricated and used a gait pacemaker, as shown in Figure 5. To minimize intersubject variability in gait speed and stride length, participants practiced twice at each gait speed prior to measurement. This device consisted of a tape light-emitting diode (LED), buzzer, and pressure sensor. Two tape LEDs were placed in parallel with a width of approximately 50 cm. The left- and right-tape LEDs indicated the contact positions of the left and right feet, respectively. The contact position was indicated in advance by a blue LED, and the subject walked along the centerline of the path, using the LED as a landmark. At the start of walking, foot release was detected using the output value of the pressure sensor. When the subject was standing, there was a load on the pressure sensor, and when the foot left the ground, the load was removed. When foot release was detected, the blue LED sequentially changed to green at a step length and time corresponding to each gait speed, and a buzzer sound (different frequencies for left and right) sounded simultaneously. The participants walked accordingly. The buzzer produces a tempo sound at the time of ground contact to control the support and swing times as much as possible. 

#### 2.6.2. Measurement

An elastic band is used to mount the MIMU to the body (Figure 6). The elastic band was of a size appropriate for the thickness of the lower limbs of the subject and was applied such that it was not too tight and did not loosen. To standardize the fixation force, the elongation of the band was set to approximately 8% of the band length (shank circumference) when worn. This ensured consistent tension across all participants and trials. The band was secured without excessive pressure or loosening. The body segment frame B was constructed by attaching reflex markers to body landmarks for 3D motion analysis using motion capture. The reflex markers were affixed to the center of hip rotation on the left and right sides, medial and lateral femoral condyles, external and internal phalanges, first metatarsal head, fifth metatarsal head, and heel. Each subject then performed three types of gait (slow, medium, and fast) three times each, for a total of nine times. The subjects started walking after standing upright with both feet together for approximately 5 s, walked 5 m, and then stopped walking. The direction of the shank segment frame relative to the MIMU was calculated from the MIMU measurement data during the standing posture before the start of each gait [55]. The three-axis acceleration, angular velocity, and magnetic field of the shank during gait were measured at a sampling rate of 100 Hz. The measured data were transmitted to a PC via Wi-Fi at the end of the gait and recorded. Simultaneously, a motion capture system (MAC3D System, Motion Analysis, Rohnert Park, CA, USA) consisting of eight cameras was used to measure the 3D gait motions at a sampling frequency of 120 Hz. An infrared trigger signal was output from the MIMU controller, received by the infrared receiver, and input to the analog signal input of the motion capture system to synchronize the MIMU and motion capture system. The MIMU and motion capture were synchronized by inputting the trigger signal into the analog signal input of the motion capture system.

#### 2.6.3. Analysis (Comparison of MIMU and MOCAP)

The sampling rate of the motion capture was 120 Hz, and it was resampled and converted to 100 Hz using the statistical software R (v 4.4.2) to facilitate comparison with the 100 Hz MIMU data. To investigate the accuracy of attitude estimation using the Madgwick filter, we compared it with attitude angle data measured by MAC3D using the calculated root mean square error (RMSE) and correlation coefficient (CC) values. The RMSE indicates the degree of error from the motion capture, and the CC indicates the similarity or identity of the two waveforms. The first and last gait cycles were excluded from the analysis, and the middle three cycles were included. In addition, to examine the effect of the MIMU position, the mean RMSE and CC values were calculated in the sagittal, anterior frontal, and horizontal planes, and a repeated-measures analysis of variance (ANOVA) was performed using the MIMU position (lateral, anterior, and medial tibial) as a factor. The Bonferroni method was used for multiple comparisons, and the threshold for statistical significance for all tests was set at *p* < 0.05.

The posture of the body segment frame relative to the sensor frame was calculated (static calibration, functional alignment method [56]), assuming that the posture while standing still was the initial posture and that the vertical direction of the body segment frame coincided with the z-axis of the Earth frame [55,57]. In addition, the motion capture frame C and shank body frame B were aligned while standing. To align the orientation (x- and y-axes) of the motion capture frame C and Earth frame E [58], the coordinate frame was rotated using Equations (7) and (8). Here, ψEC is the yaw angle between the motion capture frame and the Earth frame, and it was calculated using the Madgwick filter by aligning the MIMU with the motion capture frame.(7)qBC=qEC ⨂ qSE ⨂  qBS,(8)qEC=cosψEC2 0 0 sinψEC2.

## 3. Results

### 3.1. Results of Experiments on Muscle Contraction Effects (Experiment 1)

Table 4 shows the maximum MIMU orientation change [°] during the dorsiflexion–relaxation and plantarflexion–relaxation of the ankle joint at each MIMU position. In addition, detailed results for each subject are presented in Appendix A.The mean values for the largest and smallest errors were 4.3 ± 1.3°, 16.3 ± 5.2°, and 3.3 ± 0.6°, respectively, with the largest error observed in the anterior and medial tibia, respectively. The mean values for the lateral, anterior, and medial tibia during plantarflexion–relaxation were 5.3 ± 2.0°, 3.9 ± 1.5°, and 3.0 ± 1.1°, respectively, with the largest error being observed in the lateral and the smallest error being observed in the medial tibia. The MIMU orientation change when worn on the medial tibia was the smallest for both dorsiflexion and plantarflexion. Figure 7 shows the mean value, standard deviation, and results of a repeated-measures two-way ANOVA for the MIMU orientation change at each mounting position using the mounting position (lateral, anterior, and medial tibia) and movement (dorsiflexion and plantarflexion) as factors. The analysis of variance showed that the main effects of the mounting position (*F*(1, 5) = 21.0, *p* = 0.006) and movement (*F*(2, 10) = 49.1, *p* < 0.001) were statistically significant. In addition, because the interaction between the mounting position and movement was significant (*F*(2, 10) = 34.5, *p* = 0.001), multiple comparisons of all groups were performed. In the dorsiflexion group, the anterior position was significantly larger than those of the other mounting positions (anterior–lateral, *p* = 0.003; anterior–medial tibia, *p* = 0.003), and in the plantarflexion group, the lateral surface was significantly larger than the other mounting positions (lateral–medial tibia, *p* = 0.020; anterior–medial tibia, *p* = 0.020). In contrast, the medial tibia showed the smallest MIMU orientation change in both dorsiflexion and plantarflexion movements. Figure 8 shows the three-dimensional waveforms of the postural changes and electromyograms of the tibialis anterior and peroneus longus muscles at each MIMU mounting position. When the EMG was detected, the posture of the MIMU began to change, and when it relaxed, the waveform returned to its initial posture. Although the amount of fluctuation differed between subjects, the direction of postural fluctuation was similar. The orientation change in the horizontal plane was the largest for both dorsiflexion and plantarflexion.

### 3.2. Results of Posture Estimation Experiments During Gait (Experiment 2)

The average values of the RMSE [°] and CC in the sagittal, frontal, and horizontal planes for slow, medium, and fast gaits are listed in Table 5 and Table 6. In addition, detailed results for each subject are presented in Appendix A. The estimation accuracy can be evaluated as being higher when the RMSE is smaller and the CC is larger. The average RMSE for all gaits and planes was 2.8°, ranging from 1.1° to 5.4° in the sagittal plane, 0.8 °to 4.6° in the frontal plane, and 1.4 to 7.7° in the horizontal plane. The mean CC in all gaits and planes was 0.913, and the range of CC was 0.993–0.999 in the sagittal plane, −0.156–0.974 in the frontal plane, and 0.879–0.986 in the horizontal plane. The mean RMSE for each position was the lowest for the lateral surface (mean, 1.6°) in the sagittal plane (mean, 1.9°) and medial tibia (mean, 3.4°) in the frontal and horizontal planes. The CC was the highest for the anterior surface (mean, 0.998) in the sagittal plane and medial tibia in the frontal (mean, 0.809) and horizontal (mean, 0.963) planes. The average RMSE and CC values for the sagittal, frontal, and horizontal planes are shown in Table 7 and Figure 9, respectively. Figure 9 also shows the results of a repeated-measures one-way ANOVA, using the mounting position (lateral, anterior, and medial tibia) as the factor. The results of the ANOVA in the RMSE showed that the main effect of the mounting position (*F*(2, 30) = 3.5, *p* = 0.043) was significant. Therefore, multiple comparisons were performed. The results showed that the medial tibia was significantly smaller than the anterior tibia (*p* = 0.024) and that there were no significant differences between the other conditions (lateral–anterior: *p* = 0.074; lateral–medial tibia: *p* = 0.561). Moreover, the analysis of variance in CC showed that the main effect of the mounting position (*F*(2, 30) = 0.9, *p* = 0.361) was not significant. These results indicate the accuracy of the estimation in three dimensions. For all the gait speeds, the RMSE at the medial tibia and CC were the smallest and largest, respectively. Figure 10 shows the waveforms representing the mean and standard deviation of all shank posture data used in the analysis at each gait speed. The gait cycle was divided and standardized by identifying the initial contact (IC) timing on the sensor side from the acceleration data. In addition, detailed results for each subject are presented in Appendix A.

## 4. Discussion

### 4.1. Effects of Muscular Protuberance

We examined the orientation change of the MIMU during the plantarflexion and dorsiflexion of the ankle joint with the lower limbs immobilized in a chair-sitting position. During dorsiflexion–relaxation, the muscle group located on the anterior shank (mainly the tibialis anterior muscle) contracted; therefore, the orientation change of the MIMU worn on the anterior shank was significantly larger than that of the other mounting positions. The orientation change of the MIMU worn on the lateral shank was significantly larger than that measured in other positions. In contrast, when the MIMU was mounted on the medial tibia, the orientation change was smaller during dorsiflexion–relaxation and plantarflexion–relaxation. The MIMU posture fluctuation during plantar–dorsiflexion was primarily attributable to the uplift of the mounting surface via muscle contraction because the thigh and shank were fixed in the chair-sitting position. That is, because there is no muscle directly under the medial tibia and the effect of muscle protuberance is minimal, the medial tibia is suggested as the optimal mounting position for estimating shank posture in repetitive muscle contraction, shortening, and stretching exercises. The effects of muscle contraction on the medial, lateral, and anterior tibia were less significant.

The orientation change in the MIMU in the sagittal, frontal, and horizontal planes showed that the horizontal plane had the largest orientation change during dorsiflexion and plantarflexion, with the change exceeding 10°, which is large for posture estimation. These fluctuations are thought to cause misalignment of the body segment frame and result in estimation errors. The direction and amount of orientation change of the MIMU are thought to differ depending on whether the MIMU is positioned at the center or edge of the muscle. Therefore, it was appropriate to mount the MIMU on the medial tibia, which was less affected by muscle protrusions. The orientation change in the MIMU was larger in male subjects than in female subjects, suggesting that the amount of change differs depending on the amount of muscle mass.

These results are consistent with previous findings that soft tissue artifacts (STA), including muscle bulging and skin displacement, can significantly affect the accuracy of motion analysis via motion capture, particularly when markers are placed in active muscle groups [45,48]. Yoshida et al. (2022) quantitatively showed that muscle contraction causes the substantial displacement of markers mounted on the skin [48]. Similarly, Peters et al. (2009) reported that areas with thicker muscle tissue and higher activity were more prone to STA-related kinematic errors [45]. The findings of this study support these observations and suggest that the medial tibia is a favorable physical anatomical location for MIMU positioning when the purpose is to minimize the STA during dynamic tasks involving muscle contraction.

### 4.2. Accuracy and Validity of Posture Estimation During Gait

The posture estimation of the shank was performed for three MIMU mounting positions (lateral, anterior, and medial tibia) and three different gait types (slow, 2 km/h; medium, 3 km/h; and fast, 4 km/h). The results were compared with those obtained via motion capture. The mean RMSE for all gaits and planes in this study was 2.8°, and the mean CC was 0.913. The mean RMSE values were 2.4°, 2.2°, and 3.8° in the sagittal, frontal, and horizontal planes, respectively, and the mean CC values were 0.997, 0.786, and 0.955, respectively. McGinley et al. (2009) reported that in most common clinical situations, an error of 2° or less is likely to widely be considered as acceptable, errors between 2° and 5° are also likely to be regarded as reasonable but may require consideration during data interpretation, and errors in excess of 5° should raise concern and may be large enough to mislead clinical interpretation [8]. In addition, the majority of previous studies reported errors of 2–5°, and the estimated errors in this study were within this range. For example, Teruyama et al. (2013) estimated the shank posture angle during treadmill gait (gait speed: 3 km/h). The RMSE was reported to be approximately 2–3° in the sagittal plane, and the CC was approximately 0.995–0.998 [59]. The mean values of RMSE and CC in our study (gait speed: 3 km/h) were 2.5° and 0.998, respectively, in the sagittal plane, which is equivalent to the estimation accuracy. Furthermore, the estimation accuracy in our study is close to the RMSE of the shank posture reported by Watanabe et al. (gait speed: 3 km/h, sagittal RMSE: approximately 1.5–2.5°) [60] and Glen et al. (gait speed 4 km/h, 1.5° in the sagittal plane and 4.7° in the frontal plane) [26]. Shank posture estimation during gait often limits analysis to the sagittal plane. To the best of our knowledge, the RMSE and CC values for the horizontal plane have not been reported in previous studies. Although these are reference values, a review of lower-limb joint angle estimation reported that the RMSE values in the frontal and horizontal planes ranged from 1.0 to 6.7° and 1.4 to 6.5°, respectively [61]. Therefore, compared to the RMSE and CC values of previous studies in which the shank posture angle was calculated by MIMU and MOCAP, the estimation accuracy was equivalent, suggesting that the validity and credibility of the posture estimation in this study were sufficient.

Although no formal frequency response or drift analysis was performed, walking trials at three different speeds ensured stable signal performance under a variety of motion dynamics. No drift was observed under the conditions in this study, and only one sensor model was used to maintain consistency. We recognize that a more comprehensive evaluation, including comparisons among different sensors and long-term stability testing, is an important direction for future research.

### 4.3. Optimal MIMU Mounting Position on the Shank

As shown in Table 7, when examining the accuracy of posture estimation in three dimensions, the estimation accuracy was higher for the medial tibia (mean RMSE, 2.6°; mean CC, 0.923), lateral tibia (mean RMSE, 2.9°; mean CC, 0.917), and anterior tibia (mean RMSE, 3.0°; mean CC, 0.899). These results are similar to those of Experiment 1, which examined the effects of muscle contraction in the chair-sitting position, suggesting that the medial tibia, which was less affected by muscle protrusions, was the optimal MIMU mounting position on the shank. When the MIMU was mounted on the medial tibia, the estimation accuracy was higher than that for other mounting positions, particularly in the frontal plane. The CC was lower when the MIMU was mounted on the anterior shank, where the influence of muscle protuberance was greater, indicating that the similarity and reliability between the MIMU and motion capture-measured posture waveforms were low. Unlike high-frequency mechanical noise, soft tissue artifacts induced by muscle contractions are often nonstationary and difficult to eliminate via simple filtering. Although EMG-based correction can be explored in future studies, it introduces complexity and is sensitive to noise. Therefore, minimizing the STA through appropriate sensor placement, such as on the medial tibia, offers a more robust and practical solution for wearable and clinical applications. However, the accuracy in the sagittal plane did not differ significantly among the mounting positions. These results may be attributable to the larger range of motion in the sagittal plane and smaller range of motion in the frontal plane. Errors between the body segment frame and sensor frame owing to soft tissue artifacts, such as muscle protuberances, may cause larger kinematic crosstalk in other planes (especially in the frontal plane) because of the larger range of motion in the sagittal plane during gait, which could reduce estimation accuracy.

In addition, depending on the amount of subcutaneous fat, the medial tibia is flatter than the lateral and anterior shanks, which is thought to provide greater stability for sensor mounting. The reproducibility of the sensor position is reduced because the body parts are not flat. The standard mounting positions were along the lateral and anterior shanks, which were located on the muscles and had relatively large curvatures. Although it can be inferred that a smaller sensor is less affected by sway, it is more reasonable to select a curved surface with less curvature as the MIMU mounting position. Furthermore, the medial tibia is considered suitable in terms of not only the effect of muscle contraction but also ease of mounting. In clinical applications, the MIMU system should be easy to detach and attach, display feedback, and be wearable, usable, robust, and affordable [62]. The anterior shank of the medial tibia was used as a landmark. If the MIMU is mounted along the medial tibia and the height is marked by the tibial tuberosity, then the fit can be reproduced, and the user can mount the device by him/herself. If the ankle joint can be plantarflexed/dorsiflexed, then it can be easily identified by placing the hand on the medial tibia and checking for the presence of muscle protuberance. In clinical diagnosis, rehabilitation, and routine evaluation at home, simplicity and accuracy of measurement and setup are important. Therefore, the optimization of the MIMU mounting position in posture estimation will lead to reliable evaluations and reproducible MIMU gait analyses.

One limitation of this study was the relatively small sample size (n = 6) of healthy adult participants. Although efforts have been made to control intersubject variability through standardized instructions, gait pacing using an LED pacemaker, and pretrial practice, the limited number of subjects may have affected the generalizability of the findings. Future studies should include larger and more diverse participant cohorts to validate the results and enhance their applicability to broader populations, including clinical groups.

## 5. Conclusions

The wearable MIMU is easy to mount and dismount from the body parts, making it suitable for gait evaluation in clinics and homes. Moreover, it enables the monitoring and evaluation of various diseases, such as knee osteoarthritis, spinal cord injury, Parkinson’s disease, and hemiplegia after stroke. It is also widely used for the real-time posture control of prosthetic limbs and exoskeletal robots, FES control, orthotic selection, and braking force adjustments of short-limb orthoses. However, the optimal MIMU mounting position has seldom been studied, and the effect of the MIMU mounting position on posture-estimation accuracy is unknown. Therefore, it is necessary to clarify the effects of muscle protrusions caused by muscle contraction and shortening on posture estimation, depending on the mounting position. In this study, we analyzed shank posture estimation and conducted comparative experiments using three MIMU mounting positions: the lateral, anterior, and medial tibia, which are considered to have less influence on muscle protrusions when the anatomical body structure is considered. To investigate the optimal MIMU mounting position on the shank, we repeatedly performed plantarflexion/dorsiflexion and relaxation of the ankle joint in a chair-sitting position, and we examined the effects of muscle contraction on the posture of the MIMU (Experiment 1). We also performed posture estimation during gait and compared the 3D shank posture measured by the MIMU with that measured via motion capture to evaluate the estimation accuracy for each mounting position (Experiment 2). The results of these two experiments suggest that the medial tibia is the optimal position on the shank, as the posture estimation accuracy was the highest when the MIMU was mounted on the medial tibia, where there was no muscle under the mounting surface and the MIMU was less affected by muscle protuberance. These findings may lead to simple, reliable, and reproducible gait analysis.

These findings provide practical insights into the development of wearable motion-tracking systems, particularly for gait monitoring applications. Identifying optimal sensor mounting positions can improve the measurement accuracy, which is important for clinical assessment, rehabilitation, and home gait evaluation.

## Figures and Tables

**Figure 1 sensors-25-02273-f001:**
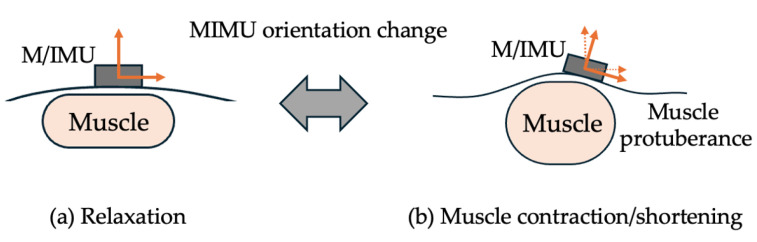
MIMU orientation change with muscle contraction. (**a**) The curvature of the body surface (MIMU mounting surface) during relaxation is small. (**b**) During muscle contraction and shortening, muscle thickness increases, and the body surface bulges (muscular protuberance), resulting in changes in MIMU posture.

**Figure 2 sensors-25-02273-f002:**
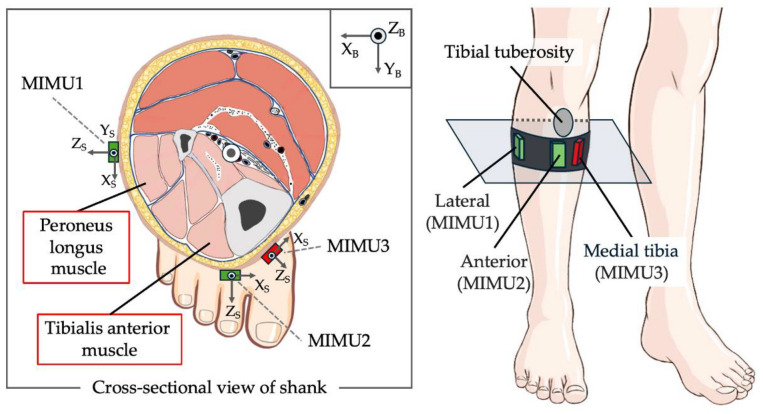
Positions of the MIMU and muscle groups present under the mounting surface. In addition to the common IMU positions on the lateral (MIMU1) and anterior shank (MIMU2), a MIMU is mounted on the medial tibia (MIMU3). The lateral aspect of the shank (MIMU1) is located on the peroneus muscle, and the anterior shank (MIMU2) is located on the tibialis anterior muscle. In contrast, there is no muscle under the mounting surface of the medial tibia (MIMU3). The X_B_, Y_B_, and Z_B_ terms indicate the orientation of the body frame B, whereas X_S_, Y_S_, and Z_S_ indicate the orientation of the sensor frame S.

**Figure 3 sensors-25-02273-f003:**
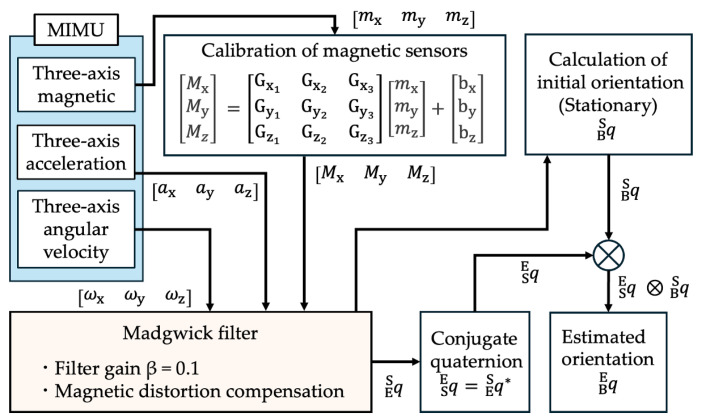
The magnetic sensor was calibrated before the measurement. Here, *m* is the magnetometer output, *M* is the corrected magnetic field, G is the sensitivity, and b is the offset. The triaxial acceleration *a*, triaxial angular velocity *w*, and corrected triaxial magnetometer *M* were input to the Madgwick filter. Then, during gait, the orientation of the Earth frame relative to the sensor frame is calculated sequentially. The initial posture qBS is calculated by the Madgwick filter from the static posture during standing, and the product of the posture estimated by the Madgwick filter with the conjugate quaternion qSE is calculated and converted into a body frame.

**Figure 4 sensors-25-02273-f004:**
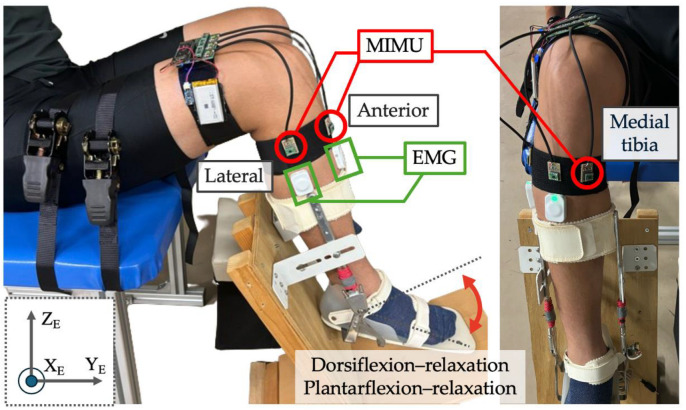
Experiment (Experiment 1) in which the ankle joint was plantarflexed/dorsiflexed in a seated posture. The MIMU was mounted to the lateral, anterior, and medial tibia of the right shank, and postural changes of the MIMU owing to muscle contraction were measured. Simultaneous EMG measurements of the tibialis anterior and peroneus longus muscles were performed to confirm that the tibialis anterior muscle was active during dorsiflexion and that the peroneus longus muscle was active during plantarflexion. Electrodes were applied along the MIMU.

**Figure 5 sensors-25-02273-f005:**
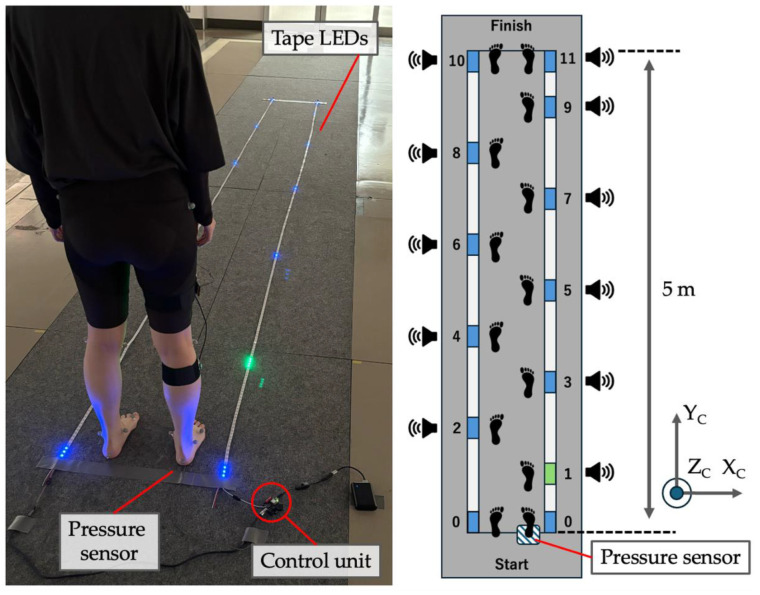
Experiment (Experiment 2) in which posture estimation during gait was performed. A homemade gait pacemaker was used to control the gait condition. This device consists of a tape LED, buzzer, and pressure sensor. A blue LED indicates the grounding position in advance, and a green LED changes to indicate the next grounding position at a preset timing. In addition, a buzzer plays a tempo sound at the timing of ground contact to control the support and swing time. The timing of the start of gait (heel release) is detected by a pressure sensor and used as a trigger for guidance. The X_C_, Y_C_, and Z_C_ terms indicate the orientation of the motion capture frame C.

**Figure 6 sensors-25-02273-f006:**
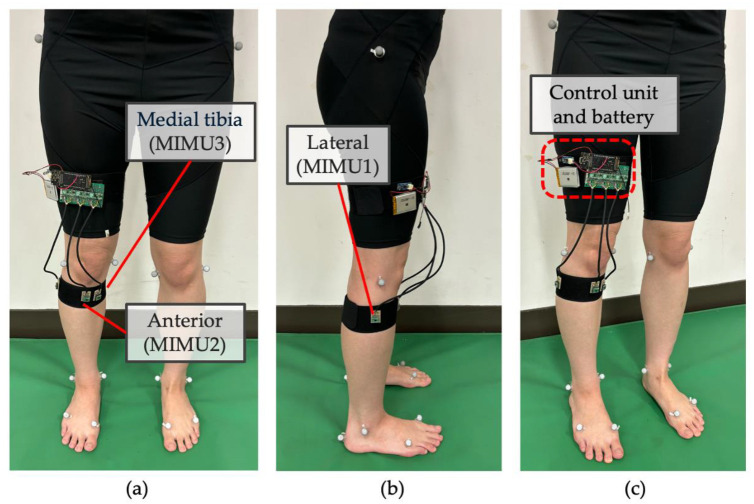
(**a**–**c**) show the front, side, and anterior oblique angles, respectively. Reflex markers were mounted on the left and right hip rotation centers, medial and lateral femoral condyles, external and internal phalanges, first metatarsal head, fifth metatarsal head, and heel of each participant to analyze gait via motion capture. MIMUs were mounted on the lateral, anterior, and medial tibiae of the right shank. In addition, a control unit and battery were mounted on the thigh.

**Figure 7 sensors-25-02273-f007:**
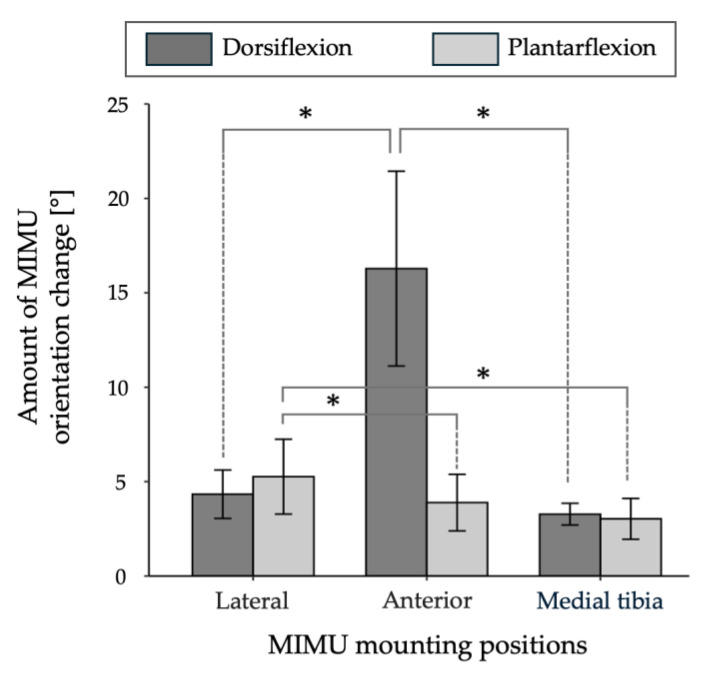
Mean value and standard deviation of the maximum MIMU orientation change at each MIMU mounting position during dorsiflexion and plantarflexion. The results of multiple comparisons after a two-way ANOVA are also shown (*: *p* < 0.05). During ankle dorsiflexion, orientation variability was significantly greater when the MIMU was mounted on the anterior shank compared with the other mounting positions. In contrast, during plantar flexion, the orientation variability was significantly greater when the MIMU was mounted on the lateral shank compared with the other mounting positions.

**Figure 8 sensors-25-02273-f008:**
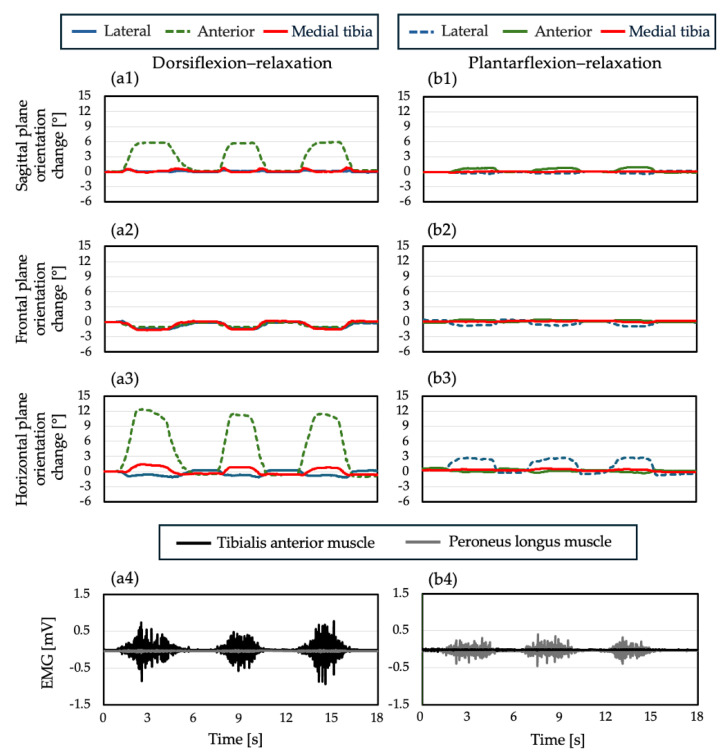
(**a1**) Sagittal plane, (**a2**) frontal plane, and (**a3**) horizontal plane, representing orientation change during dorsiflexion–relaxation. (**b1**) Sagittal plane, (**b2**) frontal plane, and (**b3**) horizontal plane, representing orientation change during plantar flexion–relaxation. The blue, green, and red lines indicate the waveforms when the device was mounted on the lateral, anterior, and medial tibia, respectively. The contraction of muscles located below the plane of mounting occurs (anterior during dorsiflexion–relaxation and lateral during plantar flexion–relaxation) and is indicated by a dotted line. Here, (**a4**) shows the electromyographic waveforms during dorsiflexion–relaxation, and (**b4**) shows those during plantarflexion–relaxation. The black and gray lines represent the anterior tibialis and peroneus longus muscles, respectively. The postural variability of the anterior MIMU is increased during tibialis anterior muscle contraction, and the postural variability of the lateral MIMU is increased during peroneus longus muscle contraction.

**Figure 9 sensors-25-02273-f009:**
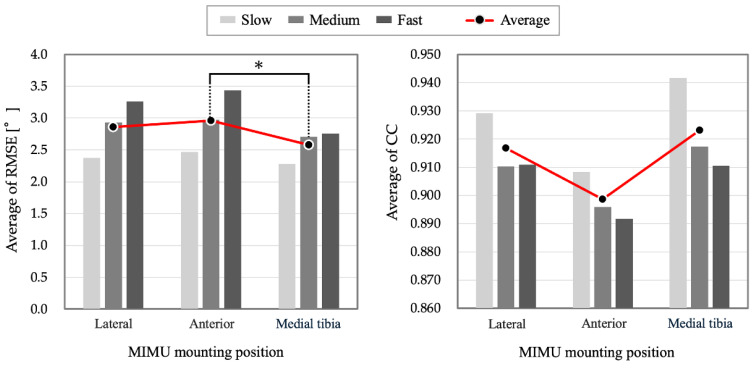
The bar graphs show the average RMSE and CC values at each gait speed, and the line graphs show the average values for all gaits at each mounting position. The results of multiple comparisons after a one-way ANOVA are also shown (*: *p* < 0.05). The mean values of the RMSE were smaller in the order of medial tibia, lateral, and anterior, and the mean values of CC were larger in the order of medial tibia, lateral, and anterior. Thus, the accuracy of posture estimation during gait was higher for the medial tibia, lateral, and anterior, in that order.

**Figure 10 sensors-25-02273-f010:**
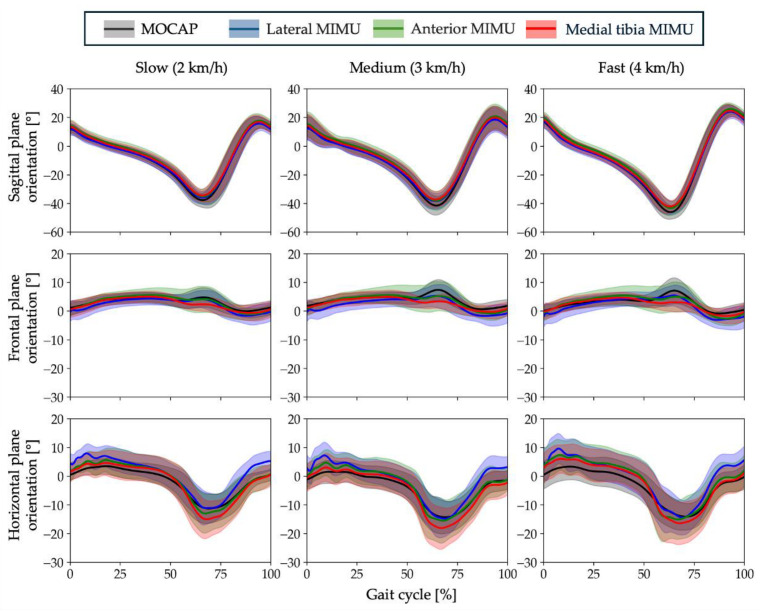
Waveforms representing the mean and standard deviation of all shank posture data used in the analysis at each gait speed. The black line indicates motion capture, the blue line indicates lateral MIMU, the green line indicates anterior MIMU, and the red line indicates medial tibia MIMU. In particular, the standard deviation of the anterior MIMU in the frontal plane was found to have a larger standard deviation compared with other mounting positions.

**Table 1 sensors-25-02273-t001:** Examples of MIMU mounting positions on the shank in previous studies that conducted posture estimations of lower limbs. A MIMU is commonly mounted on the anterior and lateral surface of the shank, although a few studies exist in which the MIMU is mounted on the medial tibia.

MIMU Mounting Position	Year	Author	Reference
Lateral	2009	Cooper G et al.	[26]
	2013	Laudanski A et al.	[27]
	2013	Kong W et al.	[28]
	2014	Palermo E et al.	[29]
	2022	Jiang C et al.	[30]
	2022	Mascia G et al.	[31]
Anterior	2007	O’Donovan KJ et al.	[32]
	2013	Tadano S et al.	[33]
	2019	Miyazaki T et al.	[34]
	2022	Rhudy MB et al.	[35]
	2024	Cornish BM et al.	[36]
Medial tibia	2019	Reenalda J et al.	[37]
	2019	Kianifar R et al.	[38]
	2022	Haji Hassani R et al.	[39]

**Table 2 sensors-25-02273-t002:** Sensor characteristics of the MIMU, which contains an accelerometer, a gyro sensor, and a magnetic sensor. The measurement ranges are ±4 g, ±1000°/s, and ±4900 μT, respectively. The accelerometer and gyro sensor were low-pass filtered with cutoff frequencies of 111 and 361 Hz, respectively.

	Accelerometer	Gyro Sensor	Magnetic Sensor [AK09916]
Measuring range	±4 g	±1000°/s	±4900 μT
Resolution	16 bit	16 bit	16 bit
Sensitivity (1 LSB)	0.12 mg	0.03°/s	0.15 μT
Noise	230 μg/√Hz	0.015°/√Hz	/
LPF	111.4 Hz	361.4 Hz	/

**Table 3 sensors-25-02273-t003:** Physical characteristics of the six subjects (three male and three female). The six subjects are represented by S1–S6. The mean age was 23 ± 1 [years], height was 1.65 ± 0.13 [m], weight was 58 ± 14 [kg], thigh length was 0.42 ± 0.04 [m], and shank length was 0.36 ± 0.03 [m].

Subject	Age [years]	Sex	Height [m]	Weight [kg]	Thigh Length [m]	Shank Length [m]
S1	23	Man	1.76	64	0.45	0.39
S2	23	Man	1.72	77	0.41	0.38
S3	22	Man	1.81	68	0.48	0.41
S4	24	Woman	1.49	41	0.39	0.34
S5	22	Woman	1.57	49	0.43	0.33
S6	22	Woman	1.53	46	0.38	0.33
Average	23 ± 1	/	1.65 ± 0.13	58 ± 14	0.42 ± 0.04	0.36 ± 0.03

**Table 4 sensors-25-02273-t004:** Mean and standard deviation [°] of the maximum orientation change of the MIMU during ankle plantarflexion/dorsiflexion, with results separated by gender. During dorsiflexion, the postural variability was the greatest when the MIMU was mounted on the anterior shank, and during plantarflexion, the postural variability was greatest when the MIMU was mounted on the lateral shank. In addition, the amount of postural variability was greater in males than in females.

	Dorsiflexion–Relaxation			Plantarflexion–Relaxation		
	Lateral	Anterior	Medial Tibia	Lateral	Anterior	Medial Tibia
Men	4.8 ± 1.6	20.0 ± 0.9	3.7 ± 0.1	6.7 ± 1.6	4.4 ± 1.2	3.5 ± 1.2
Women	3.8 ± 0.9	12.6 ± 4.9	2.8 ± 0.5	3.9 ± 1.1	3.4 ± 1.8	2.5 ± 0.9
All	4.3 ± 1.3	16.3 ± 5.2	3.3 ± 0.6	5.3 ± 2.0	3.9 ± 1.5	3.0 ± 1.1

**Table 5 sensors-25-02273-t005:** Average RMSE value [°] for each gait, mounting position, and each analysis plane. The MIMU mounting position with the lowest RMSE for each gait speed in each plane is indicated by shading. In the sagittal plane, the lateral MIMU had the smallest RMSE (mean 2.1 [°]), and in the frontal and horizontal planes, the medial tibial MIMU had the smallest RMSE (mean 1.9 [°] and 3.4 [°], respectively).

	Sagittal Plane			Frontal Plane			Horizontal Plane		
	Lateral	Anterior	Medial Tibia	Lateral	Anterior	Medial Tibia	Lateral	Anterior	Medial Tibia
Slow	1.7	2.5	2.3	1.9	2.0	1.5	3.5	2.9	3.0
Medium	2.2	2.8	2.6	2.5	2.5	2.0	4.1	3.7	3.6
Fast	2.3	3.0	2.5	2.7	2.7	2.1	4.8	4.6	3.7
All	2.1	2.7	2.4	2.4	2.4	1.9	4.1	3.7	3.4

The MIMU mounting position with the lowest RMSE for each gait speed in each plane is indicated by shading.

**Table 6 sensors-25-02273-t006:** Average CC for each gait, mounting position, and analysis plane. The MIMU mounting position with the largest CC for each gait speed in each plane is indicated by shading. In the sagittal plane, the anterior MIMU had the largest CC (mean 0.998), and in the frontal and horizontal planes, the medial tibial MIMU had the largest CC (mean 0.809 and 0.963, respectively).

	Sagittal Plane			Frontal Plane			Horizontal Plane		
	Lateral	Anterior	Medial Tibia	Lateral	Anterior	Medial Tibia	Lateral	Anterior	Medial Tibia
Slow	0.997	0.998	0.997	0.837	0.780	0.863	0.953	0.947	0.965
Medium	0.997	0.998	0.997	0.779	0.732	0.794	0.955	0.957	0.961
Fast	0.997	0.998	0.997	0.793	0.725	0.771	0.943	0.953	0.963
All	0.997	0.998	0.997	0.803	0.746	0.809	0.950	0.952	0.963

The MIMU mounting position with the largest CC for each gait speed in each plane is indicated by shading.

**Table 7 sensors-25-02273-t007:** Mean RMSE and CC values in the sagittal, frontal, and horizontal planes calculated for each gait speed. The shaded area indicates the MIMU position with the highest estimation accuracy in three dimensions. For all gait speeds, the average RMSE was smaller, and the average CC was larger when the MIMU was mounted on the medial tibia. These results suggest that the medial tibia is the most accurate location for 3D posture estimation.

	Average RMSE (Sagittal, Frontal, and Horizontal Planes)			Average CC (Sagittal, Frontal, and Horizontal Planes)		
	Lateral	Anterior	Medial Tibia	Lateral	Anterior	Medial Tibia
Slow	2.4	2.5	2.3	0.929	0.908	0.942
Medium	2.9	3.0	2.7	0.910	0.896	0.917
Fast	3.3	3.4	2.8	0.911	0.892	0.910
All	2.9	3.0	2.6	0.917	0.899	0.923

The shaded area indicates the MIMU position with the highest estimation accuracy in three dimensions.

## Data Availability

The original contributions of this study are included in the article and Appendix A.

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
