# Peer review of "Optimization of MIMU Mounting Position on Shank in Posture Estimation Considering Muscle Protuberance"

_sensors, 2025, doi:10.3390/s25072273_

Round 1

Reviewer 1 Report

Comments and Suggestions for Authors

The authors explored optimizing the mounting position of magnetic/inertial measurement units (M/IMU) on the shank to assess posture accuracy. They investigated ankle joint plantar/dorsiflexion and relaxation in a chair-sitting position, along with how muscle contraction affects device posture. This manuscript is interesting, well-prepared, and will capture a wide readership interested in gait analysis and monitoring. And hence, it would be accepted in Sensors with minor revisions.

Comments:

  • The abbreviations of M/IMU and MIMU are potentially confusing. It is advisable to modify them to ensure clear distinction.
  • It is suggested to add a table comparing the advantages of the proposed system for demonstrating the novelty of this work.

Author Response

We would like to thank the reviewers for their valuable comments and suggestions.

Comments 1: The abbreviations of M/IMU and MIMU are potentially confusing. It is advisable to modify them to ensure clear distinction.

Response 1: Thank you for the comment. We agree that using both “M/IMU” and “MIMU” could lead to confusion for readers. To improve clarity and consistency, we have revised the manuscript and unified the terminology under “MIMU,” which refers to magnetic-inertial measurement units. The definition of “MIMU” is now clearly stated at its first appearance, and the term is used consistently throughout the revised manuscript.

Comments 2: It is suggested to add a table comparing the advantages of the proposed system for demonstrating the novelty of this work.

Response 2: Thank you for the suggestion. We understand the value of including a comparative table to highlight novelty. The novelty of this study lies in the development of a new posture estimation method rather than in the experimental design and focus. Specifically, this study uniquely investigated the influence of MIMU sensor mounting positions under identical gait conditions and separately evaluated the effect of muscle contraction on sensor accuracy through a seated dorsiflexion/plantarflexion experiment. Previous studies have often focused on sensor algorithm development or single-location evaluations; however, our approach focuses on independent factors affecting posture estimation by isolating sensor placement and muscular influence. We have clarified this point more explicitly in the revised Introduction section to better communicate the novelty of our approach.

  The following sentences have been added on page 5, paragraph 1, lines 9–14 of the revised manuscript: “Although many previous studies have focused on improving posture-estimation algorithms or comparing different sensor systems, the present study offers a novel perspective by comparing MIMU mounting positions under identical movement conditions and experimentally examining the influences of muscle contraction on the estimation error. This dual approach provides new insights into the optimization of sensor placement for MIMU-based gait analyses.”.

Reviewer 2 Report

Comments and Suggestions for Authors

Here, the authors present an important study on optimizing M/IMU mounting positions on the shank for accurate posture estimation while considering muscle protuberance. It provides insightful comparisons among different sensor placements and evaluates their impact on estimation accuracy. The methodology is well-structured, and the experimental approach using plantar/dorsiflexion tests and gait analysis with optical motion capture is understandable. However, there are certain drawbacks and areas requiring improvement before the manuscript can be considered for publication.

Recommendation: Major Revision Required

Some Comments

  1. The comparison among different mounting positions in the manuscript is clear, statistical significance (e.g., p-values, confidence intervals) is not sufficiently addressed. Please include statistical tests such as ANOVA or t-tests would enhance the rigor of the findings.
  2. The authors mention that soft tissue artifacts influence sensor accuracy, but there is limited discussion on mitigating these effects. The inclusion of filtering techniques or sensor fusion approaches would strengthen the study.
  3.  The manuscript primarily evaluates accuracy based on optical motion capture validation, but other characterization methods could enhance reliability. For example, frequency response analysis of sensor signals to determine signal quality across different movement speeds. Drift analysis over time to assess stability of M/IMU sensors in different positions. Comparative analysis of different sensor brands/models to ensure generalizability of results.
  4. The details regarding subject recruitment and variability in test subjects are insufficient.
  5. It would be helpful to include a more detailed description of how movement variability was controlled among subjects.
  6. Captions should provide a more comprehensive description of the content rather than relying solely on figure numbers in the text.
  7. The abstract should explicitly mention the key findings (e.g., "Accuracy improved by X% when mounting on the medial tibia").
  8. The conclusion should be more forward-looking, discussing how these findings can be applied in practical wearable sensor systems or clinical settings.

Comments on the Quality of English Language

The English could be improved to more clearly express the research.

Author Response

We sincerely appreciate the constructive feedback provided by the reviewers. We had the manuscript proofread by a professional English editor, including the revised and newly added sentences after the peer review.

Comments 1: The comparison among different mounting positions in the manuscript is clear, statistical significance (e.g., p-values, confidence intervals) is not sufficiently addressed. Please include statistical tests such as ANOVA or t-tests would enhance the rigor of the findings.

Response 1: Thank you for your helpful comment regarding the statistical analysis. In Experiment 1 (ankle dorsiflexion/plantar flexion in a seated posture), we performed a two-way ANOVA to evaluate the effects of the sensor position and direction of motion on the measurement accuracy. In Experiment 2 (gait-based posture estimation), accuracy was evaluated using the RMSE and correlation coefficient (CC) as standard measurement criteria, following the methodologies of previous studies. Because this evaluation metric compares signal similarities between MIMU and motion capture data, we did not believe that statistical tests such as ANOVA were necessary. However, in response to your suggestion, we performed a one-way ANOVA with the MIMU wearing position as a factor.

  The following sentences have been added on page 12, paragraph 1, lines 8–13 of the revised manuscript: “In addition, to examine the effect of the MIMU position, the mean RMSE and CC values were calculated in the sagittal, anterior frontal, and horizontal planes, and a repeated-measures analysis of variance (ANOVA) was performed, using the MIMU position (lateral, anterior, medial tibial) as a factor. The Bonferroni method was used for multiple comparisons, and the threshold for statistical significance for all tests was set at p < 0.05.”.

  The following sentences have been added on page 15, paragraph 1, lines 5–12 of the revised manuscript: “Figure 9 also shows the results of a repeated-measures one-way ANOVA, using the mounting position (lateral, anterior, and medial tibia) as the factor. The results of the ANOVA in the RMSE showed that the main effect of the mounting position (F(2, 30) = 3.5, p = 0.043) was significant. Therefore, multiple comparisons were performed. The results showed that the medial tibia was significantly smaller than the anterior tibia (p = 0.024) and that there were no significant differences between the other conditions (lateral–anterior: p = 0.074; lateral–medial tibia: p = 0.561). Moreover, the analysis of variance in CC showed that the main effect of the mounting position (F(2, 30) = 0.9, p = 0.361) was not significant.”.

Comments 2: The authors mention that soft tissue artifacts influence sensor accuracy, but there is limited discussion on mitigating these effects. The inclusion of filtering techniques or sensor fusion approaches would strengthen the study.

Response 2: We agree that soft tissue artifacts (STA) can adversely affect the accuracy of the acceleration and angular velocity signals. As noted in the revised Discussion section, STAs, particularly those induced by muscle contractions, are often nonstationary in nature, making them difficult to remove using conventional filtering techniques. One possible approach to mitigate an STA is to monitor muscle activity using electromyography (EMG) and apply signal-dependent corrections. However, EMG sensors are highly susceptible to noise and are not always practical in clinical or wearable applications in which minimizing the number of sensors is preferable. Given these constraints, we considered the anatomical factors and selected the medial aspect of the tibia as the sensor mounting location. This region is less affected by muscle contractions during gait and is therefore less prone to STA. We believe that this is a practical and clinically reasonable strategy for improving the measurement reliability without the use of additional hardware.

  The following sentences have been added on page 19, paragraph 2, lines 12–17 of the revised manuscript: “Unlike high-frequency mechanical noise, soft tissue artifacts induced by muscle contractions are often nonstationary and difficult to eliminate via simple filtering. Although EMG-based correction can be explored in future studies, it introduces complexity and is sensitive to noise. Therefore, minimizing the STA through appropriate sensor placement, such as on the medial tibia, offers a more robust and practical solution for wearable and clinical applications.”.

Comments 3: The manuscript primarily evaluates accuracy based on optical motion capture validation, but other characterization methods could enhance reliability. For example, frequency response analysis of sensor signals to determine signal quality across different movement speeds. Drift analysis over time to assess stability of M/IMU sensors in different positions. Comparative analysis of different sensor brands/models to ensure generalizability of results.

Response 3: We agree that such an analysis could improve the reliability and generalizability of our findings. In this study, we evaluated gait at three different speeds (slow, medium, and fast), which allowed us to observe signal behavior under varying dynamic conditions. The results were consistent across speeds, indicating a stable signal quality. No noticeable drift was observed under the conditions of this study, and only one sensor model was used to ensure consistency in the data collection. However, owing to the limited revision period of 10 days, we were unable to implement the suggested additional analyses. We recognize the importance of these methods and plan to address them in future studies to further strengthen the robustness and applicability of our approach.

  The following sentences have been added on page 19, paragraph 1, lines 1–6 of the revised manuscript: “Although no formal frequency response or drift analysis was performed, walking trials at three different speeds ensured stable signal performance under a variety of motion dynamics. No drift was observed under the conditions in this study, and only one sensor model was used to maintain consistency. We recognize that a more comprehensive evaluation, including comparisons among different sensors and long-term stability testing, is an important direction for future research.”.

Comments 4: The details regarding subject recruitment and variability in test subjects are insufficient.

Response 4: We agree that a clear description of subject recruitment and intersubject variability is important for transparency and reproducibility. In the revised manuscript, we have added more detailed information regarding participant recruitment and characteristics.

  The following sentence has been added on page 8, paragraph 3, lines 2–3 of the revised manuscript: “All subjects were recruited from Waseda University.”

Comments 5: It would be helpful to include a more detailed description of how movement variability was controlled among subjects.

Response 5: Agree. In the revised manuscript, we have provided a more detailed description of the measures taken to minimize intersubject movement variability. Specifically, an LED-based gait pacemaker was used to control the temporal and spatial parameters of gait in all participants to ensure that walking speed and step timing were consistent across the trials. Before the actual measurements, the participants performed two practice trials at each of the three designated walking speeds (slow, medium, and fast) to familiarize themselves with the task and minimize variability owing to unfamiliarity.

  The following sentence has been added on page 10, paragraph 1, lines 6–7 of the revised manuscript: “To minimize intersubject variability in gait speed and stride length, participants practiced twice at each gait speed prior to measurement.”.

Comments 6: Captions should provide a more comprehensive description of the content rather than relying solely on figure numbers in the text.

Response 6: Agree. In the revised manuscript, we have updated all figure captions to include a more comprehensive explanation of the content, including the key variables and statistical information where relevant.

  The following sentence has been added to the caption of Figure 1 on page 4: “MIMU orientation variation with muscle contraction.”.

Figure 2 on page 5: “Positions of the MIMU and muscle groups present under the mounting surface.”.

Figure 3 on page 7: “Then, during gait, the orientation of the Earth frame relative to the sensor frame is calculated sequentially. The initial posture  is calculated by the Madgwick filter from the static posture during standing, and the product of the posture estimated by the Madgwick filter with the conjugate quaternion  is calculated and converted into a body frame.”.

Figure 4 on page 9: “Experiment (Experiment 1) in which the ankle joint was plantarflexed/dorsiflexed in a seated posture.”.

Figure 5 on page 10: “Experiment (Experiment 2) in which posture estimation during gait was performed. A homemade gait pacemaker was used to control the gait condition.”.

Figure 6 on page 11: “In addition, a control unit and battery were mounted on the thigh.”.

Figure 7 on page 13: “During ankle dorsiflexion, orientation variability was significantly greater when the MIMU was mounted on the anterior shank compared with the other mounting positions. In contrast, during plantar flexion, the orientation variability was significantly greater when the MIMU was mounted on the lateral shank compared with the other mounting positions.”.

Figure 8 on page 14: “The postural variability of the anterior MIMU is increased during tibialis anterior muscle contraction, and the postural variability of the lateral MIMU is increased during peroneus longus muscle contraction.”.

Figure 9 on page 16: “The results of multiple comparisons after a one-way ANOVA are also shown (*: p < 0.05). The mean values of the RMSE were smaller in the order of medial tibia, lateral, and anterior, and the mean values of CC were larger in the order of medial tibia, lateral, and anterior. Thus, the accuracy of posture estimation during gait was higher for the medial tibia, lateral, and anterior, in that order.”.

Figure 10 on page 17: “In particular, the standard deviation of the anterior MIMU in the frontal plane was found to have a larger standard deviation compared with other mounting positions.”.

Table 1 on page 3: “Examples of MIMU mounting positions on the shank in previous studies that conducted posture estimations of lower limbs. A MIMU is commonly mounted on the anterior and lateral surface of the shank, although a few studies exist in which the MIMU is mounted on the medial tibia.”.

Table 2 on page 6: “Sensor characteristics of the MIMU, which contains an accelerometer, a gyro sensor, and a magnetic sensor. The measurement ranges are ± 4 g, ± 1000 °/s, and ± 4900 μT, respectively. The accelerometer and gyro sensor were low-pass filtered with cutoff frequencies of 111 and 361 Hz, respectively.”.

Table 3 on page 8: “Physical characteristics of the six subjects (three male and three female). The six subjects are represented by S1–S6. The mean age was 23 ± 1 [yr], height was 1.65 ± 0.13 [m], weight was 58 ± 14 [kg], thigh length was 0.42 ± 0.04 [m], and shank length was 0.36 ± 0.03 [m].”.

Table 4 on page 13: “Mean and standard deviation [°] of the maximum orientation variability of the MIMU during ankle plantarflexion/dorsiflexion, with results separated by gender. During dorsiflexion, the postural variability was the greatest when the MIMU was mounted on the anterior shank, and during plantarflexion, the postural variability was greatest when the MIMU was mounted on the lateral shank. In addition, the amount of postural variability was greater in males than in females.”.

Table 5 on page 15: “In the sagittal plane, the lateral MIMU had the smallest RMSE (mean 2.1 [°]), and in the frontal and horizontal planes, the medial tibial MIMU had the smallest RMSE (mean 1.9 [°] and 3.4 [°], respectively).”.

Table 6 on page 15: “In the sagittal plane, the anterior MIMU had the largest CC (mean 0.998), and in the frontal and horizontal planes, the medial tibial MIMU had the largest CC (mean 0.809 and 0.963, respectively).”.

Comments 7: The abstract should explicitly mention the key findings (e.g., Accuracy improved by X% when mounting on the medial tibia "").

Response 7: Agree. We have revised the abstract to explicitly state the key quantitative findings of this study. Specifically, we reported the RMSE and CC values for each MIMU mounting position and highlighted that the medial tibia yielded the highest accuracy.

  The following sentences have been added on page 19, paragraph 1, lines 1–6 of the revised manuscript: “In Experiment 1, the posture variation at the medial tibia was significantly smaller than that at the other positions, showing an 80% reduction compared with the anterior tibia during dorsiflexion. In Experiment 2, the medial tibia achieved the highest estimation accuracy, showing a 13% lower RMSE than that of the anterior position.”.

Comments 8: The conclusion should be more forward-looking, discussing how these findings can be applied in practical wearable sensor systems or clinical settings.

Response 8: Agree. In the revised manuscript, we have expanded the Conclusion section to include a more forward-looking discussion. Specifically, we discuss how the identified optimal sensor placement can inform the design of wearable gait-monitoring systems, improve sensor placement guidelines in clinical assessments, and support the development of real-time feedback tools in rehabilitation settings.

  The following sentences have been added on page 20, paragraph 2, lines 1–4 of the revised manuscript: “These findings provide practical insights into the development of wearable motion-tracking systems, particularly for gait-monitoring applications. Identifying optimal sensor mounting positions can improve the measurement accuracy, which is important for clinical assessment, rehabilitation, and home gait evaluation.” .

Reviewer 3 Report

Comments and Suggestions for Authors

The manuscript titled "Optimization of M/IMU mounting position on shank in posture estimation considering muscle protuberance," investigates the effect of mounting magnetic/inertial measurement units (M/IMU) at different positions on the shank for posture estimation accuracy, with a focus on the impact of muscle protuberance on measurement errors. The study experimentally validates the superiority of the inner tibia as the M/IMU mounting position, providing valuable optimization suggestions for applications such as gait analysis, rehabilitation monitoring, and prosthetic control. The overall structure of the manuscript is clear, the experimental design is reasonable, the analysis methods are appropriate, and the research conclusions are of practical value. However, there are still issues that can be improved:

1.In Section 1.3, the manuscript mentioned the influence of muscle eminence on sensor posture, but did not discuss the influence of the presence of more extensive soft tissue artifacts (STA) on sensor posture. STA is an important topic in gait estimation, and there is considerable literature on this. It is recommended to include a more detailed discussion of STA and cite relevant literature to strengthen the theoretical background of the manuscript.

  1. In Section 2.4, the manuscriptmentions the use of 6 subjects (3 male and 3 female), which is a small sample size that may affect the generalizability of the research results. It is suggested to include a discussion of this limitation in the subsequent discussion section or provide an explanation as to whether the sample size is sufficient.

3.In Section 2.6.1, the manuscript mentions the use of a gait pacemaker to control gait speed and stride length, but the description of the experimental device is somewhat brief. More details about the methods used with this device can be provided, or an explanation can be included as to whether its effectiveness has been validated in other studies.

  1. In Section 2.6.2, the manuscriptstates: "An elastic band was used to attach the M/IMU to the body. The elastic band was of a size appropriate for the thickness of the lower limbs of the subject and was applied such that it was not too tight and did not loosen." The tightness of the band could affect measurement accuracy. It is recommended to add standardized methods for sensor fixation and discuss how it may influence the errors.
  2. The mechanism should be discussed in details based on experimental results and reference.
  3. The advantages of manuscript should be stated clearly in the introduction part.

Author Response

We are grateful to the reviewers for their thoughtful and detailed reviews, which helped us improve the manuscript.

Comments 1: In Section 1.3, the manuscript mentioned the influence of muscle eminence on sensor posture, but did not discuss the influence of the presence of more extensive soft tissue artifacts (STA) on sensor posture. STA is an important topic in gait estimation, and there is considerable literature on this. It is recommended to include a more detailed discussion of STA and cite relevant literature to strengthen the theoretical background of the manuscript.

Response 1: Thank you for your valuable comment. We agree that soft tissue artifacts (STAs) present crucial issues in human motion analysis and warrant a more detailed discussion. In the original manuscript, we briefly mentioned the influence of muscle eminence on sensor posture; however, we have now expanded Section 1.3 to include a broader discussion of STAs, including both muscle deformation and skin displacement.

The following sentences have been added on page 4, paragraph 2, lines 13–25 of the revised manuscript: “In addition to the effects of muscle expansion, soft tissue artifacts (STAs), including skin displacement and soft tissue deformation, are widely recognized as major sources of error in human motion analyses. STAs result from the relative motion between skin, muscle, and underlying bone during dynamic activity and can significantly distort kinematic measurements. Potentially, motion-capture STA can be a major source of distortion of kinematic measurements. Several studies have quantified the STA when optical motion capture markers are attached to the body and have shown that they introduce inaccuracies in kinematic calculations [45–49]. In addition, compensation methods have been proposed for optical motion-capture systems [50,51]; however, such methods are not applicable to wearable IMU-based cases. Thus, a more realistic and clinically relevant approach to gait analysis using an IMU would be to minimize the effects of the STA by selecting an anatomically stable mounting position, such as the medial tibia.”.

  1. Peters A, Galna B, Sangeux M, Morris M, Baker R. Quantification of soft tissue artifact in lower limb human motion analysis: a systematic review. Gait Posture. 2010 Jan;31(1):1-8. doi: 10.1016/j.gaitpost.2009.09.004.
  2. Fiorentino NM, Atkins PR, Kutschke MJ, Goebel JM, Foreman KB, Anderson AE. Soft tissue artifact causes significant errors in the calculation of joint angles and range of motion at the hip. Gait Posture. 2017 Jun;55:184-190. doi: 10.1016/j.gaitpost.2017.03.033.
  3. D'Isidoro F, Brockmann C, Ferguson SJ. Effects of the soft tissue artefact on the hip joint kinematics during unrestricted activities of daily living. J Biomech. 2020 May 7;104:109717. doi: 10.1016/j.jbiomech.2020.109717.
  4. Yoshida Y, Matsumura N, Yamada Y, Yamada M, Yokoyama Y, Miyamoto A, Nakamura M, Nagura T, Jinzaki M. Three-Dimensional Quantitative Evaluation of the Scapular Skin Marker Movements in the Upright Posture. Sensors (Basel). 2022 Aug 29;22(17):6502. doi: 10.3390/s22176502.
  5. Wang Y, Guo J, Tang H, Li X, Guo S, Tian Q. Quantification of soft tissue artifacts using CT registration and subject-specific multibody modeling. J Biomech. 2024 Jan;162:111893. doi: 10.1016/j.jbiomech.2023.111893.
  6. Lahkar BK, Rohan PY, Assi A, Pillet H, Bonnet X, Thoreux P, Skalli W. Development and evaluation of a new methodology for Soft Tissue Artifact compensation in the lower limb. J Biomech. 2021 Jun 9;122:110464. doi: 10.1016/j.jbiomech.2021.110464.
  7. Einfeldt A, Budde L, Ortigas-Vásquez A, Sauer A, Utz M, Jakubowitz E. A new method called MiKneeSoTA to minimize knee soft-tissue artifacts in kinematic analysis. Scientific Reports. 2024; 14: 20666. https://doi.org/10.1038/s41598-024-71409-z.

Comments 2: In Section 2.4, the manuscript mentions the use of 6 subjects (3 male and 3 female), which is a small sample size that may affect the generalizability of the research results. It is suggested to include a discussion of this limitation in the subsequent discussion section or provide an explanation as to whether the sample size is sufficient.

Response 2: Agree. The study limitations regarding sample size are noted in the Discussion section.

The following sentences have been added on page 19, paragraph 4, lines 1–7 of the revised manuscript: “One limitation of this study was the relatively small sample size (n = 6) of healthy adult participants. Although efforts have been made to control intersubject variability through standardized instructions, gait pacing using an LED pacemaker, and pretrial practice, the limited number of subjects may have affected the generalizability of the findings. Future studies should include larger and more diverse participant cohorts to validate the results and enhance their applicability to broader populations, including clinical groups.”.

Comments 3: In Section 2.6.1, the manuscript mentions the use of a gait pacemaker to control gait speed and stride length, but the description of the experimental device is somewhat brief. More details about the methods used with this device can be provided, or an explanation can be included as to whether its effectiveness has been validated in other studies.

Response 3: Agree. The following description of the gait pacemaker using tape LEDs has been added on page 10, paragraph 1, lines 9–18 of the revised manuscript: “Two tape LEDs were placed in parallel with a width of approximately 50 cm. The left- and right-tape LEDs indicated the contact positions of the left and right feet, respectively. The contact position was indicated in advance by a blue LED, and the subject walked along the centerline of the path, using the LED as a landmark. At the start of walking, foot release was detected using the output value of the pressure sensor. When the subject was standing, there was a load on the pressure sensor, and when the foot left the ground, the load was removed. When foot release was detected, the blue LED sequentially changed to green at a step length and time corresponding to each gait speed, and a buzzer sound (different frequencies for left and right) sounded simultaneously. The participants walked accordingly.”.

Comments 4: In Section 2.6.2, the manuscript states: "An elastic band was used to attach the M/IMU to the body. The elastic band was of a size appropriate for the thickness of the lower limbs of the subject and was applied such that it was not too tight and did not loosen." The tightness of the band could affect measurement accuracy. It is recommended to add standardized methods for sensor fixation and discuss how it may influence the errors.

Response 4: Agree. The following description for the elastic band fixing force has been added on page 11, paragraph 1, lines 3–6 of the revised manuscript: To standardize the fixation force, the elongation of the band was set to approximately 8% of the band length (shank circumference) when worn. This ensured consistent tension across all participants and trials. The band was secured without excessive pressure or loosening.”.

Comments 5: The mechanism should be discussed in detail based on experimental results and reference.

Response 5: Agree. We have expanded the Discussion section to include a more detailed explanation of the underlying mechanism based on both the experimental findings and relevant literature. We have expanded the Discussion section to include a more detailed explanation of the mechanism based on both experimental findings and relevant literature.

The following sentences have been added on page 18, paragraph 2, lines 1–10 of the revised manuscript: “These results are consistent with previous findings that soft tissue artifacts (STA), including muscle bulging and skin displacement, can significantly affect the accuracy of motion analysis via motion capture, particularly when markers are placed in active muscle groups [45,48]. Yoshida et al. (2022) quantitatively showed that muscle contraction causes the substantial displacement of markers mounted on the skin [48]. Similarly, Peters et al. (2009) reported that areas with thicker muscle tissue and higher activity were more prone to STA-related kinematic errors [45]. The findings of this study support these observations and suggest that the medial tibia is a favorable physical anatomical location for MIMU positioning when the purpose is to minimize the STA during dynamic tasks involving muscle contraction.”.

Comments 6: The advantages of manuscript should be stated clearly in the introduction part.

Response 6: Agree. We have revised the Introduction section to explicitly highlight the main advantages of our work. We emphasize the following. 1. This study focused on a systematic comparison of multiple MIMU mounting positions on a shank under identical gait conditions, which has not been sufficiently addressed in previous research. 2. In addition to the gait analysis, a separate experiment was conducted to isolate the influence of muscle contraction on MIMU accuracy, which is an aspect that has often been overlooked in previous studies.

The following sentences have been added on page 5, paragraph 1, lines 9–14 of the revised manuscript: “Although many previous studies have focused on improving posture-estimation algorithms or comparing different sensor systems, the present study offers a novel perspective by comparing MIMU mounting positions under identical movement conditions and experimentally examining the influences of muscle contraction on the estimation error. This dual approach provides new insights into the optimization of sensor placement for MIMU-based gait analyses.

Round 2

Reviewer 2 Report

Comments and Suggestions for Authors

Accept in present form

Reviewer 3 Report

Comments and Suggestions for Authors

Accept it as it is